# BIDIRECTIONAL LANGUAGE MODELS ARE ALSO FEW-SHOT LEARNERS

**Ajay Patel** *
University of Pennsylvania

**Bryan Li**
University of Pennsylvania

**Mohammad Sadegh Rasooli**
Microsoft

**Noah Constant**
Google Research

**Colin Raffel**
UNC Chapel Hill

**Chris Callison-Burch**
University of Pennsylvania

## ABSTRACT

Large language models such as GPT-3 (Brown et al., 2020) can perform arbitrary tasks without undergoing fine-tuning after being prompted with only a few labeled examples. An arbitrary task can be reformulated as a natural language prompt, and a language model can be asked to generate the completion, indirectly performing the task in a paradigm known as prompt-based learning. To date, emergent prompt-based learning capabilities have mainly been demonstrated for unidirectional language models. However, bidirectional language models pre-trained on denoising objectives such as masked language modeling produce stronger learned representations for transfer learning. This motivates the possibility of prompting bidirectional models, but their pre-training objectives have made them largely incompatible with the existing prompting paradigm. We present SAP (Sequential Autoregressive Prompting), a technique that enables the prompting of bidirectional models. Utilizing the machine translation task as a case study, we prompt the bidirectional mT5 model (Xue et al., 2021) with SAP and demonstrate its few-shot and zero-shot translations outperform the few-shot translations of unidirectional models like GPT-3 and XGLM (Lin et al., 2021), despite mT5's approximately 50% fewer parameters. We further show SAP is effective on question answering and summarization. For the first time, our results demonstrate prompt-based learning is an emergent property of a broader class of language models, rather than only unidirectional models.

## 1 INTRODUCTION

Recent work on GPT-2 (Radford et al., 2019) and GPT-3 (Brown et al., 2020) have shown that large language models possess few-shot learning capabilities and zero-shot instruction following capabilities, despite only being pre-trained with a self-supervised causal language modeling objective (which is to predict the next token).

An arbitrary task can be converted into a natural language task specification, often called a *prompt*. Prompting a task in this way makes its format similar to the language modeling objective used to pre-train large language models. In the zero-shot setting, this prompt contains just the task with instructions, whereas in the few-shot setting, the prompt contains both the task and several example demonstrations. When a language model is tasked to generate text to complete this prompt, it can perform the task in the process. The broader paradigm of reframing all tasks as text generation is known as *prompt-based learning*. In the few-shot setting, the learning that occurs from examples provided in a given prompt (the context) is known as *in-context learning* (Liu et al., 2021). In the zero-shot setting, models perform *instruction following* (Ouyang et al., 2022), with their performance guided through natural language instructions provided in the prompt.

Emergent prompt-based learning capabilities have mainly been demonstrated for unidirectional language models. Bidirectional language models have stronger learned representations (Devlin et al., 2019; Conneau et al., 2020; Raffel et al., 2020); however, they have not been able to broadly

---

*Correspondence to: `ajayp@upenn.edu`

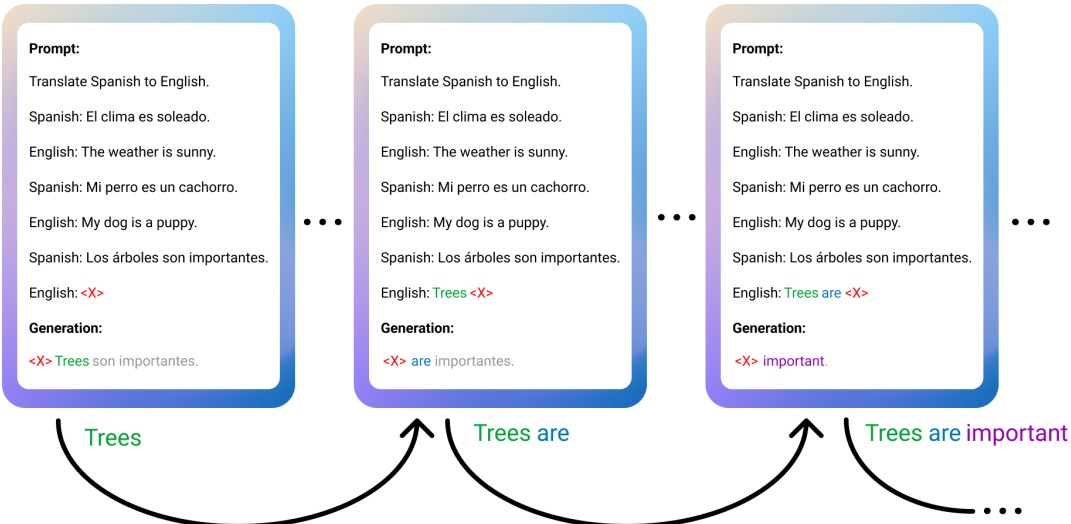

Figure 1: A visualization of our SAP technique extracting high-quality translations from mT5. In the zero-shot setting, the examples used in the prompt are synthetic examples retrieved in a fully unsupervised manner.

demonstrate the same few-shot in-context learning capabilities or zero-shot instruction following capabilities due to the incompatibility bidirectional denoising pre-training objectives have with the prompting paradigm. Instead, they typically require fine-tuning. Bidirectional models are not able to generate long, fluent completions to prompts since they are usually only trained to output single tokens or short spans of text to in-fill masked tokens during pre-training. We discuss this more in-depth in Section 2.1.

Today, language model architects are faced with a difficult choice between unidirectional or bidirectional models. The authors of GPT-3 lay out this design dilemma in Brown et al. (2020):

> "GPT-3 has several structural and algorithmic limitations ... as a result our experiments do not include any bidirectional architectures or other training objectives such as denoising ... our design decision comes at the cost of potentially worse performance on tasks which empirically benefit from bidirectionality ... making a bidirectional model at the scale of GPT-3, and/or trying to make bidirectional models work with few- or zero-shot learning, is a promising direction for future research, and could help achieve the 'best of both worlds'."

In this paper, we directly address this dilemma. We contribute a new technique, SAP (**S**equential **A**utoregressive **P**rompting), that enables bidirectional language models to take advantage of prompting and allows them to perform at the level of unidirectional models in few- or zero-shot learning without fine-tuning. SAP iteratively prompts bidirectional models, concatenating previous generations back into the prompt, to produce longer generations from models that were only pre-trained to output short, mask-infill spans. We acknowledge efficiency concerns in Section 6 and we discuss the importance and impact of SAP and its results to the field regardless of those concerns.

Using the machine translation task as an in-depth case study, we empirically demonstrate mT5 (Xue et al., 2021), a bidirectional language model, used with SAP outperforms its unidirectional counterparts, GPT-3 and XGLM (Brown et al., 2020; Lin et al., 2021) in both the few-shot and zero-shot settings, while utilizing approximately 50% fewer parameters. We then examine SAP's effectiveness on other tasks such as question answering and summarization, demonstrating that bidirectional models can be prompted for tasks beyond machine translation.

Our work hints at the possibility of more efficient and performant few-shot learners through pre-trained language models that incorporate bidirectionality. We discuss this impact and outline future research directions to this end in Section 6. In summary, our key contributions are:

1. We introduce SAP, a technique that enables bidirectional language models to work with few-shot and zero-shot prompt-based learning at a level that exceeds unidirectional models. Our results demonstrate in-context learning and instruction following are emergent properties of a broader class of language models, rather than only unidirectional models, addressing a long-standing challenge in language model design and use.

2. We perform an in-depth study of the effectiveness of a bidirectional language model, mT5, with SAP on the machine translation task. Evaluating over 14 language pairs, despite using approximately 50% fewer parameters than GPT-3 and XGLM, we find SAP with mT5 has improved average few-shot and zero-shot performance over all language pairs, and especially has improved performance on individual low-resource language pairs.

3. We propose a range of improvements—filtering, prompt ensembling, and English-centric bootstrapping—to the unsupervised machine translation procedure outlined by Han et al. (2021) to better adapt the bootstrapping process for unsupervised low-resource machine translation.

4. We assess SAP's performance on the tasks of question answering and summarization, and we find the technique enables few-shot in-context learning and zero-shot instruction following capabilities of bidirectional models in tasks beyond machine translation.

## 2 RELATED WORK

### 2.1 UNIDIRECTIONAL AND BIDIRECTIONAL LANGUAGE MODELS

Transformer-based language models (Vaswani et al., 2017) can be broadly categorized into bidirectional and unidirectional models. Bidirectional models are models that use a denoising pre-training objective (such as masked language modeling), allowing them to utilize *bidirectional* context when learning language representations. Unidirectional language models are models with a causal—or a left-to-right—language modeling objective (such as next token prediction), restricting them to be *unidirectional* when learning representations (Liu et al., 2021).

The T5 family of models, such as T5 v1.1 and mT5, and BART-style models (Lewis et al., 2019) are bidirectional, while GPT-style models, such as GPT-2, GPT-3, and XGLM are unidirectional. Usually, but not always, bidirectional models are paired with an encoder-decoder architecture, while unidirectional models are paired with a decoder-only architecture (Devlin et al., 2019; Raffel et al., 2020; Xue et al., 2021; Radford et al., 2019; Brown et al., 2020; Lin et al., 2021; Wang et al., 2022). BERT-style models are an example of an exception. BERT-style models are bidirectional, but they cannot be easily utilized for prompting and text generation since they are encoder-only (Wang & Cho, 2019). Of the available bidirectional models, T5 models are the only models with a long enough sequence length (unlimited with their relative position embeddings) to support many in-context prompt examples and with a large enough number of parameters to be effective zero-shot and few-shot performers (Radford et al., 2019; Brown et al., 2020; Kaplan et al., 2020). See Appendix J for a survey of popular open source language models. Aside from sequence length and model size, BART is not purely trained on the span denoising objective SAP exploits, but is also trained on many other corruption objectives like "Sentence Permutation." For this reason, we utilize the T5 models for experiments and leave the exploration of the generalization of SAP to other models, that could become available later, as future work.

Devlin et al. (2019) and Raffel et al. (2020) have both shown that after transfer learning, bidirectional denoising pre-training objectives such as BERT's masked language modeling and T5's random span corruption outperform causal language modeling on downstream tasks. Brown et al. (2020) concedes this to be a potential source of weakness for the GPT-3 model on certain tasks where bidirectionality is important.

Despite the advantages of denoising objectives, prompting and in-context learning capabilities have not been broadly demonstrated for bidirectional language models like T5, disqualifying them when few-shot in-context learning and zero-shot instruction following is desired. Lester et al. (2021) explains this may be because:

> "...a T5 model pre-trained exclusively on span corruption, such as T5.1.1, has never seen truly natural input text (free of sentinel tokens), nor has it ever been asked to predict truly natural targets"

In other words: when pre-trained on their denoising objectives, language models like T5 that utilize bidirectionality are only conditioned to output a single token or short spans of tokens (the in-fill of the mask) rather than full and complete sentences; this inhibits their ability to generate arbitrary-length natural responses to a variety of prompts.

Despite the stronger learned representations of bidirectional models, their shortcomings in prompt-based learning motivate Brown et al. (2020) and Lin et al. (2021) to explicitly choose unidirectional models over bidirectional models for GPT-3 and XGLM.

## 2.2 PROMPTING BIDIRECTIONAL LANGUAGE MODELS

Unlike prior approaches to incorporate prompt-based learning capabilities into bidirectional models, our technique, SAP, neither requires fine-tuning, weight updates, nor supervised instruction-tuning datasets. It demonstrates that bidirectional language models develop *innate* few-shot learning capabilities with in-context learning and zero-shot instruction following capabilities.

**Cloze-style prompts**    Schick & Schütze (2021a) and Schick & Schütze (2021b) find that bidirectional models such as RoBERTa and ALBERT (Liu et al., 2019; Lan et al., 2019) can be "prompted" with cloze-style phrases. They propose a few-shot training paradigm called PET where the model's predicted mask in-fill, called a "verbalizer," is used to label fine-tuning examples for the model. These verbalizers are only a single word or a few words, e.g. "yes", "no", "amazing", "worse". Ni & Kao (2022) follow a similar technique, but with the ELECTRA model (Clark et al., 2020). These works primarily demonstrate zero-shot effectiveness on classification tasks such as sentiment analysis, rather than more challenging generation tasks such as machine translation or question answering. Furthermore, they still require fine-tuning for effective few-shot learning, a major limitation that does not achieve the prompt-based in-context learning or instruction following abilities of unidirectional models such as GPT-3.

**LM-adaptation**    Lester et al. (2021) finds some success with prompting the T5 v1.1 models after continued pre-training on the unidirectional prefix-LM objective described in Raffel et al. (2020). The resulting model, T5 v1.1 LM-adapted (T5+LM), is described as a late-stage adaptation to a unidirectional objective. Adaptation requires performing weight updates, and given that representations learned by the original denoising objective have been shown to be superior (Raffel et al., 2020), we hypothesize that such an adaptation could degrade the quality of the learned representations.

**Prompt-tuning**    Lester et al. (2021) and Li & Liang (2021) find by fine-tuning only a portion of the parameters in an otherwise frozen pre-trained bidirectional language model, a "soft prompt" can be discovered through backpropagation. Soft prompts are prompts discovered in the embedding space of the model and are not grounded in natural language. As a form of parameter-efficient fine-tuning (Liu et al., 2022), this approach requires training the prompt embeddings and benefits from initialization from LM-adaptation, both of which require performing weight updates. The nature of soft prompts lacking grounding in natural language makes their use and flexibility limited, a stark difference from the instruction following capabilities of unidirectional models (Liu et al., 2021).

**Instruction-tuning**    Language models can be fine-tuned on a supervised dataset consisting of natural language prompts and their respective target completions (Wei et al., 2021; Sanh et al., 2022; Ouyang et al., 2022; Min et al., 2021). This "instruction-tuning" technique allows these models to improve performance on instruction following and therefore exhibit few-shot and zero-shot capabilities through prompting. The T0 model in particular is an instruction-tuned version of the T5+LM model (Lester et al., 2021), augmenting it with prompting capabilities. While instruction-tuning likely bolsters the instruction following performance of a model, we hypothesize that by instruction-tuning, the T0 model is to some degree surfacing the innate prompting ability that the bidirectional model already has. We provide evidence towards this hypothesis by demonstrating that bidirectional models can be prompted without instruction-tuning.

| | English-Russian | Russian-English |
|---|---|---|
| **Prompting (mT5$_{3.7B}$)** | | |
| Using the full generation from the first time step only – $G_0$ | 1.9 | 5.6 |
| **Sequential Prompting (mT5$_{3.7B}$ + SP)** | | |
| Concatenating the full generation at each time step – CONCAT$(G_0, ..., G_{t-1})$ | 9.3 | 17.9 |
| **Sequential Autoregressive Prompting (mT5$_{3.7B}$ + SAP)** | | |
| Concatenating the first word of the generation at each time step – CONCAT$(F_0, ..., F_{t-1})$ | **20.1** | **26.9** |

Table 1: Few-shot (2-shot) machine translation results on FLORES-101 devtest (spBLEU) using mT5$_{3.7B}$ as described in Section 3. In this experiment, over two language pairs, English-Russian and Russian-English, we compare a) simply prompting the model once and taking the full generation $G_0$ b) concatenating the full generation at each time step $G_t$ to the prompt in the next time step c) concatenating just the first word of the generation at each time step $F_t$ to the prompt in the next time step.

### 2.3 UNSUPERVISED MACHINE TRANSLATION THROUGH PROMPTING

GPT-2 (Radford et al., 2019) and GPT-3 (Brown et al., 2020) have shown it is possible to perform few-shot machine translation and unsupervised zero-shot machine translation with large language models using prompting and in-context learning. The XGLM model (Lin et al., 2021) trains a similar architecture to GPT-3 on a diverse multilingual corpus, resulting in improvements on few-shot, low-resource machine translation. Han et al. (2021) introduce bootstrapping and self-amplification techniques to further improve unsupervised zero-shot performance on machine translation.

## 3 FEW-SHOT MACHINE TRANSLATION

To motivate our method for enabling few-shot in-context learning in bidirectional language models, we first focus on applying mT5$_{3.7B}$ (mT5-XL) (Xue et al., 2021) to the machine translation task as an in-depth case study since this task benefits greatly from bidirectionality (Conneau et al., 2020; Lin et al., 2021). We largely follow the procedure of Lin et al. (2021), except with mT5 and SAP. mT5 is a massively multilingual bidirectional model trained on random span corruption, a variant of masked language modeling. We demonstrate that with SAP, mT5 can perform few-shot machine translation using prompting and in-context examples with no fine-tuning. We first formulate a prompt format that utilizes its random span masking scheme to complete the translation task, such as:

```
Translate Spanish to English.
Spanish: El clima es soleado.
English: The weather is sunny.
Spanish: Mi perro es un cachorro.
English: My dog is a puppy.
Spanish: Los árboles son importantes.
English: <X>
```

### 3.1 SEQUENTIAL AUTOREGRESSIVE PROMPTING (SAP) TECHNIQUE

By requiring mT5 to in-fill <X>[1], we are effectively asking it to translate the Spanish sentence. However, due to the limitations of the denoising pre-training objective on prompting (described in Section 2.1), we observe mT5 often outputs a partial translation of the beginning of the source sentence, rather than the full translation. To overcome this, we prompt mT5 $T$ times until the model generates a stop token [2], resulting in a longer translation. At each time step of iteration, we keep the first word generated (using the space character as delimiter) and concatenate it into the last line of the prompt to use in the next time step. This iterative prompting enables us to extract longer generations. Formally, we denote the generation at each time step $t$ as $G_t$. We denote the first word generated at each time step as $F_t$, where $F_t = $ SPLIT$(G_t, $ " " $)$ [0]. We update the prompt at each time step $P_t$ to include the cumulative generation from all previous time steps concatenated in the

---

[1]We use the first sentinel token from the mT5 vocabulary as our mask token.
[2]We repurpose the 100th sentinel token from the mT5 vocabulary as our stop token.

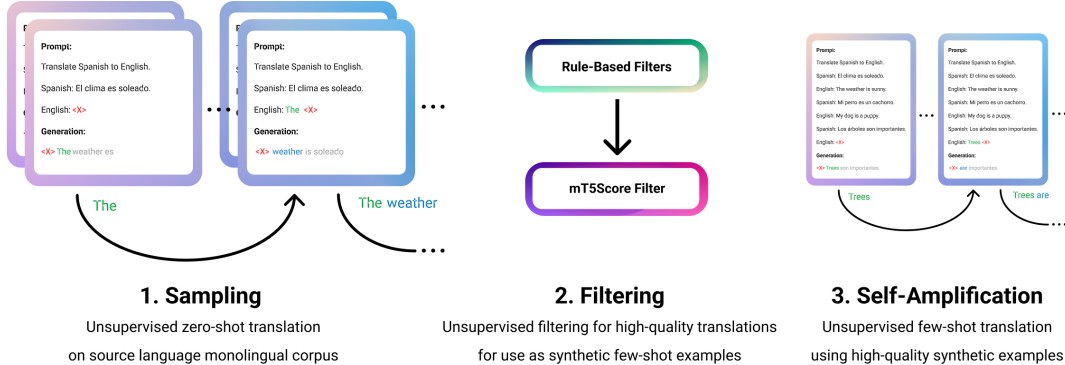

Figure 2: A visualization of the bootstrapping process described in Section 4.

last line of the prompt. The prompt used at each time step $P_t$ is as follows:

> Translate Spanish to English.
> Spanish: El clima es soleado.
> English: The weather is sunny.
> Spanish: Mi perro es un cachorro.
> English: My dog is a puppy.
> Spanish: Los árboles son importantes.
> English: CONCAT($F_0, \ldots, F_{t-1}$) <X>

In Table 1, we also consider sequential prompting—concatenating the entire generation $G_t$ instead of just the first word of the generation $F_t$—but find that it produces significantly inferior results as low-quality tokens are generated after the first word. By conditioning the model to generate the next word in the translation based on previous words generated, this technique resembles autoregression. mT5 is already autoregressive, but it is autoregressive only at the decoder level. Adding previously generated words back into the prompt allows them to pass through the encoder layers as well. For this reason, we call this technique SAP (**S**equential **A**utoregressive **P**rompting). To provide a signal to stop generation, we add our stop token at the end of each example in the prompt. We stop prompting after the model generates a stop token.[3] The overall process is graphically depicted, with stop tokens omitted, in Figure 1.

## 3.2 RESULTS

Following Lin et al. (2021), we evaluate our technique on 14 languages from the FLORES-101 dataset (Goyal et al., 2021) that span high-resource and low-resource languages[4]. We evaluate SentencePiece BLEU (spBLEU) (Goyal et al., 2021) in every direction, leading to an evaluation over 182 language pairs in total. Abbreviated results can be found in Table 2, and the matrix of full results can be found in Appendix A. Examples generations can be found in Appendix K.

On an average spBLEU score over all 182 pairs, our model matches the performance of the unidirectional XGLM and GPT-3 models—with approximately 50% fewer parameters and 16x fewer examples. Notably, our technique has significant improved performance on language pairs with at least one low-resource language, while trailing only slightly on high-resource pairs.

## 4 UNSUPERVISED ZERO-SHOT MACHINE TRANSLATION

To extend our in-depth case study on the machine translation task, we now perform fully unsupervised zero-shot machine translation with SAP and mT5 following the procedure of Han et al. (2021), which uses a self-amplification technique to boost performance. A comparison of zero-shot

---

[3]We also implement a basic post-processing step to strip any generated text after a repeated sequence of three or more tokens following settings available in common decoding implementations (Wolf et al., 2019).

[4]HR: English (en), German (de), French (fr), Catalan (ca), Finish (fi), Russian (ru), Bulgarian (bg), Chinese (zh), Korean (ko), Arabic (ar), Swahili (sw); LR: Hindi (hi), Malayalam (my), Tamil (ta)

performance without self-amplification can be found in Appendix G. We ultimately will replace the examples in the few-shot prompt with synthetic parallel examples. These synthetic parallel examples are bootstrapped in a completely unsupervised fashion using a zero-shot translation prompt with no examples. The zero-shot prompt format looks like:

> Translate Spanish to English.
> Spanish: Los árboles son importantes.
> English: <X>

We adapt the bootstrap process of Han et al. (2021) to retrieve these synthetic parallel examples. The process, as depicted in Figure 2, consists of three steps:

**Step 1 (sampling)**: Generate synthetic parallel examples using a zero-shot translation prompt (with no examples) to translate sentences from a monolingual source language corpus.

**Step 2 (filtering)**: Filter out low-quality synthetic examples to keep only high-quality synthetic examples using an unsupervised scoring technique (discussed in Section 4.1).

**Step 3 (self-amplification)**: Translate any source language sentence desired using these synthetic parallel examples in the few-shot prompt.

We iteratively run multiple rounds of this bootstrap by repeating step 2 and step 3 to form a better few-shot prompt. The few-shot prompt after self-amplification is used to translate more source language sentences. These are then filtered using the scoring technique used in step 2 and so on. In our experiments, we run four bootstrapping rounds and sample 100 source language sentences from the training dataset in each round. Note that none of the target language parallel sentences from the training dataset are used in this zero-shot setting; following Han et al. (2021), only the source language sentences are used.

### 4.1 FILTERING DOWN TO HIGH-QUALITY TRANSLATIONS

The filtering step of the bootstrap requires an unsupervised scoring method for assessing the quality of translations. We first use `langdetect`[5], a language identifier, as a simple rule-based filter to ensure the generated text is in the desired target language. We then score the remaining generated translations against their corresponding original sentence in the source language. For this unsupervised multilingual similarity metric, we utilize the BERTScore (Zhang et al., 2019) algorithm with $mT5_{300M}$ (mT5-small)[6], dubbing it "mT5Score". We ablate the use of mT5Score as a filter in Appendix C.

We take the top two synthetic parallel examples with the highest mT5Score in the filtering step and use those as synthetic few-shot examples in the prompt in the self-amplification step.

### 4.2 TRANSLATING WITH AN ENSEMBLE OF PROMPTS

Because the two examples used in the prompt can greatly affect the quality of the generated translations, some prompts containing low-quality synthetic examples may cause poor translations for certain sentences. To combat this and reduce variation in performance, we keep the top $N$ synthetic examples instead of two synthetic examples. We use these to form $\frac{N}{2}$ different few-shot prompts with two synthetic parallel examples each. Each sentence in the test set is then translated with these $\frac{N}{2}$ different prompts to produce $\frac{N}{2}$ translations. The best translation of the $\frac{N}{2}$ translations is chosen in a fully unsupervised manner with mT5Score, as done in the filtering step of the bootstrap.

We find this ensembling technique helps make unsupervised zero-shot performance competitive with few-shot performance. Experiments varying the number of prompts in the ensemble can be found in Appendix D. Unless otherwise stated, we use a 4 prompt ensemble in this paper: $\frac{N}{2} = 4$.

In sum, we sample and zero-shot translate 100 sentences from a monolingual corpus, keep the top eight synthetic parallel examples scored by mT5Score, and use them to form four few-shot prompts, each of which has two synthetic examples.

---

[5] https://pypi.org/project/langdetect/
[6] The BERTScore Python library (Zhang et al., 2019) directly supports using mT5 instead of BERT.

| | | HR → HR | LR → HR | HR → LR | LR → LR | All |
|---|---|---|---|---|---|---|
| Number of Language Pairs | | 110 | 33 | 33 | 6 | 182 |
| Supervised | | 21.5 | 10.3 | 8.6 | 4.3 | 16.6 |
| GPT-3$_{6.7B}$ | (32-shot) | 8.1 | 0.4 | 0.1 | 0.1 | 5.0 |
| XGLM$_{7.5B}$ | (32-shot) | 15.3 | 8.7 | 6.8 | 3.8 | 12.2 |
| mT5$_{3.7B}$ + SAP | (2-shot) | 14.5 | 9.8 | 8.2 | 7.1 | 12.3 |
| mT5$_{3.7B}$ + SAP | (zero-shot) | **15.5** | **10.7** | **9.1** | **8.2** | **13.2** |

Table 2: Abbreviated few-shot and unsupervised zero-shot machine translation results on FLORES-101 devtest (spBLEU). The matrix of full results can be found in Appendix A. Results are average spBLEU scores over subsets of the 182 language pairs (`src → tgt`) where "LR" is a low-resource language and "HR" is a high-resource language. "All" represents the average spBLEU score over all 182 language pairs. Supervised results correspond to the M2M-124 615M model from Goyal et al. (2021). XGLM$_{7.5B}$ results correspond to the model from Lin et al. (2021). Bold denotes best of GPT-3, XGLM, and mT5. spBLEU computed using the implementation from Goyal et al. (2021).

### 4.3 ENGLISH-CENTRIC BOOTSTRAPPING

While Han et al. (2021) only performed a bootstrap on English-French and French-English pairs, we perform bootstrapping on some language pairs which may contain at least one low-resource language or non-English language.

It has been found that multilingual language models perform best in English, due to imbalance of languages in the pre-training corpus where English has the highest amount of data (Lin et al., 2021). Therefore, when running the bootstrap on various language pairs, we modify the bootstrap to favor generating English, or pivot through English when neither the source nor target language is English. Ablation experiments can be found in Appendix E. We outline examples of our modified English-centric bootstrapping process for various language pairs in Appendix F.

### 4.4 RESULTS

We report results with the same method used for our few-shot evaluation. Abbreviated results can be found in Table 2 and the matrix of full results can be found in Appendix A.

In this unsupervised setting, we find our zero-shot results exceed our 2-shot results; furthermore, they significantly exceed the performance of the XGLM and GPT-3 results reported in Lin et al. (2021) on an average spBLEU score over all 182 pairs (+1.0 spBLEU). Again, we note strong performance on language pairs that contain one or more low-resource languages.

Intuitively, we can explain the zero-shot performance surpassing the few-shot performance through our use of prompt ensembling in the zero-shot setting. As prompt ensembling utilizes four prompts with two synthetic parallel examples each, it essentially uses eight synthetic examples, instead of just two real examples in the few-shot setting. Our synthetic examples are nearly as high-quality as real examples (similar to the findings of Han et al. (2021)) as demonstrated by Appendix D. Prompt ensembling not only reduces performance variation if low-quality synthetic examples are selected during the bootstrap, but it also boosts performance beyond the few-shot setting as demonstrated by Table 1 and the Appendix D experiments (Russian-English 26.9 → 27.9 spBLEU).

In Appendix B, we also evaluate on WMT14 (Bojar et al., 2014) to compare with the results reported in Han et al. (2021) using GPT-3$_{175B}$.

## 5 OTHER LANGUAGE GENERATION TASKS

We next demonstrate that bidirectional models have a generalized ability, beyond machine translation, to be prompted for arbitrary tasks. We evaluate their performance on question answering and summarization language generation tasks. Example generations can be found in Appendix K.

| | | en | ar | de | el | es | hi | ru | th | tr | vi | zh | avg |
|---|---|---|---|---|---|---|---|---|---|---|---|---|---|
| XGLM$_{7.5B}$ | (zero-shot) | 19.5/31.9 | 12.9/29.6 | 12.2/25.3 | 7.2/28.2 | 12.5/24.0 | **11.0**/14.0 | 10.9/27.8 | **16.8/26.4** | 13.6/26.8 | 12.5/21.2 | 13.2/20.3 | 12.9/25.0 |
| mT5$_{3.7B}$ + SAP | (zero-shot) | **25.0/48.8** | **17.4/39.4** | **19.4/43.0** | **9.7/41.0** | **15.0/42.1** | 6.6/**32.1** | **16.1/39.0** | 2.8/17.4 | **15.8/37.0** | **18.2/41.9** | **15.0/29.0** | **14.6/37.3** |

Table 3: Zero-shot multilingual question answering results (EM/F1) on the XQuAD test set (Artetxe et al., 2020).

| | | EM | F1 |
|---|---|---|---|
| *Zero-shot* | | | |
| T5+LM$_{3B}$ | (zero-shot) | 23.5 | 48.4 |
| mT5$_{3.7B}$ + SAP | (zero-shot) | 30.2 | 54.0 |
| *Few-shot* | | | |
| mT5$_{3.7B}$ | (16-shot) | 23.0 | 54.5 |
| mT5$_{3.7B}$ + SAP | (16-shot) | **35.4** | **60.0** |

Table 4: Zero-shot and few-shot question answering results on the SQuAD v1.1 dev set (Rajpurkar et al., 2016).

| | | ROUGE-1 | ROUGE-2 | ROUGE-L |
|---|---|---|---|---|
| *Zero-shot* | | | | |
| T5+LM$_{3B}$ | (zero-shot) | 5.3 | 0.6 | 4.9 |
| mT5$_{3.7B}$ | (zero-shot) | 15.4 | 4.6 | 14.5 |
| mT5$_{3.7B}$ + SAP | (zero-shot) | **22.0** | **7.4** | **20.1** |
| *Few-shot* | | | | |
| T5+LM$_{3B}$ | (2-shot) | 14.1 | 4.4 | 13.2 |
| mT5$_{3.7B}$ | (2-shot) | 15.9 | 4.5 | 15.0 |
| mT5$_{3.7B}$ + SAP | (2-shot) | **22.0** | **6.8** | **20.3** |

Table 5: Zero-shot and few-shot summarization results on the CNN / Daily Mail v3.0.0 test set evaluated with ROUGE (Nallapati et al., 2016; See et al., 2017; Hermann et al., 2015; Lin, 2004).

## 5.1 QUESTION ANSWERING

We compare the zero-shot question answering performance of mT5 against XGLM on the XQuAD dataset (Artetxe et al., 2020), a multilingual question answering dataset, in Table 3. We find mT5 with SAP outperforms XGLM significantly (+1.7 EM, +12.3 F1).

In Table 4, we also compare against T5+LM (Lester et al., 2021). As T5+LM is English-only, we compare using the English-only SQuAD v1.1 dataset (Rajpurkar et al., 2016). We still utilize the multilingual mT5 with SAP due to observations that the English-only T5 v1.1 model does not perform as well as mT5 in prompt-based learning[7]. SAP achieves +6.7 EM and +5.6 F1 over T5+LM.

SAP, as an iterative technique, is useful for producing long generations from a bidirectional model for tasks such as machine translation. We find, however, it still has utility on tasks like question answering where answer generations are shorter spans of text. We ablate utilizing SAP with mT5 against the simple approach of prompting mT5 once and using the mask in-fill generated on SQuAD v1.1. In the few-shot (16-shot) setting, we find that utilizing SAP still markedly improves performance (+12.5 EM, +5.5 F1) even on short-form generation tasks like question answering.

## 5.2 SUMMARIZATION

We next perform summarization on the CNN/Daily Mail dataset (Nallapati et al., 2016; See et al., 2017; Hermann et al., 2015) as another long-form text generation task. We compare mT5 with T5+LM and ablate the usage of SAP once again in Table 5. In the few-shot setting, we find a significant lead against T5+LM (+7.1 ROUGE-L). Of that +7.1 ROUGE-L boost, the ablation of our usage of SAP finds the technique itself is responsible for a large component of the boost (+5.3).

## 6 CONCLUSION AND FUTURE DIRECTIONS

We demonstrate SAP with the bidirectional mT5 model enables few-shot and zero-shot machine translation and zero-shot multilingual question answering, outperforming unidirectional models despite using far fewer parameters and examples. Our results suggest that the bidirectional representations learned by models such as mT5 contribute to this improved performance. Still, we concede that our results do not conclusively prove bidirectionality explains the difference in performance. Beyond bidirectionality and pre-training objectives, mT5, XGLM, and GPT-3 further differ in architecture, pre-training corpus, and hyperparameters. A complete ablation experiment would be

---

[7]We discuss this observation in more detail in Appendix H.

computationally expensive, and we leave this as future work. The main limitation of SAP lies in its computational efficiency, discussed further in Appendix I along with potential mitigations.

Importantly, these results demonstrate bidirectional models possess few-shot in-context learning and zero-shot instruction following capabilities innately, without the post-hoc modifications required by prior work. Our results finally contribute strong evidence towards the strength and efficiency of bidirectional pre-training objectives and motivate further research into bidirectional architectures, pre-training objectives, and language models designed and optimized for prompting and few-shot learning. We hypothesize these future bidirectional training schemes could yield an approach that overcomes the efficiency limitations of SAP, while maintaining the performance and parameter size reduction benefits. Concurrent recent work that compares or mixes unidirectional and bidirectional pre-training objectives (Wang et al., 2022; Tay et al., 2022; Soltan et al., 2022) already provide some early evidence towards this hypothesis.

ACKNOWLEDGMENTS

We thank Daphne Ippolito for reviewing versions of this draft and Victor Sanh for answering queries related to earlier directions of this work. This research is based upon work supported in part by the DARPA KAIROS Program (contract FA8750-19-2-1004), the DARPA LwLL Program (contract FA8750-19-2-0201), the IARPA BETTER Program (contract 2019-19051600004), the IARPA HIA-TUS Program (contract 2022-22072200005), and the NSF (Award 1928631). Approved for Public Release, Distribution Unlimited. The views and conclusions contained herein are those of the authors and should not be interpreted as necessarily representing the official policies, either expressed or implied, of ODNI, DARPA, IARPA, NSF, or the U.S. Government.

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

# A  FLORES-101 Few-shot and Unsupervised Zero-shot Machine Translation

| | | | en | de | fr | ca | fi | ru | bg | zh | ko | ar | sw | hi | my | ta | avg |
|---|---|---|---|---|---|---|---|---|---|---|---|---|---|---|---|---|---|
| en | Supervised | | – | 32.6 | 42.0 | 31.2 | 24.2 | 27.1 | 37.4 | 19.3 | 18.5 | 17.9 | 26.9 | 28.1 | 3.5 | 3.4 | 24.0 |
| | GPT-3$_{6.7B}$ | (32-shot) | – | 25.9 | **36.1** | 23.8 | 10.2 | 11.2 | 5.9 | 12.5 | 1.2 | 1.1 | 0.5 | 0.3 | 0.1 | 0.0 | 9.9 |
| | XGLM$_{7.5B}$ | (32-shot) | – | **27.6** | 36.0 | **34.0** | **23.3** | **24.2** | **33.1** | **15.6** | **12.0** | 11.5 | 18.0 | **19.9** | 11.0 | 8.5 | **21.1** |
| | mT5$_{3.7B}$ + Sap | (2-shot) | – | 23.2 | 34.2 | 26.2 | 15.8 | 20.1 | 27.9 | 9.5 | 10.4 | 11.4 | 17.3 | 14.0 | 11.0 | 11.2 | 17.9 |
| | mT5$_{3.7B}$ + Sap | (zero-shot) | – | 26.0 | 33.2 | 28.4 | 15.7 | 21.2 | 27.1 | 11.3 | 10.5 | **12.7** | **19.1** | 16.1 | **13.2** | **13.1** | 19.0 |
| de | Supervised | | 35.8 | – | 35.5 | 25.8 | 22.6 | 24.6 | 31.5 | 17.2 | 16.6 | 14.8 | 21.0 | 23.4 | 2.3 | 2.3 | 21.0 |
| | GPT-3$_{6.7B}$ | (32-shot) | 40.4 | – | 26.2 | 17.2 | 8.1 | 9.3 | 4.8 | 9.0 | 1.0 | 0.9 | 0.5 | 0.3 | 0.1 | 0.1 | 9.1 |
| | XGLM$_{7.5B}$ | (32-shot) | 38.8 | – | **27.9** | 19.1 | **20.5** | **19.7** | **25.8** | **12.3** | 3.4 | 6.6 | 11.7 | **14.3** | 9.9 | 4.8 | 16.5 |
| | mT5$_{3.7B}$ + Sap | (2-shot) | 33.0 | – | 24.4 | 17.8 | 14.1 | 15.7 | 20.2 | 8.2 | **9.1** | 7.7 | 11.0 | 10.0 | 9.8 | 9.6 | 14.7 |
| | mT5$_{3.7B}$ + Sap | (zero-shot) | 35.9 | – | 25.9 | **22.5** | 14.3 | 17.4 | 21.0 | 8.2 | 8.4 | 8.7 | 13.4 | 10.4 | 9.0 | **10.8** | 15.8 |
| fr | Supervised | | 37.2 | 28.5 | – | 28.7 | 21.9 | 24.5 | 32.2 | 17.6 | 16.7 | 15.4 | 17.2 | 22.9 | 2.1 | 0.8 | 20.4 |
| | GPT-3$_{6.7B}$ | (32-shot) | 42.8 | 20.9 | – | 23.7 | 8.0 | 9.7 | 4.6 | 9.1 | 1.0 | 1.0 | 0.4 | 0.3 | 0.1 | 0.0 | 9.4 |
| | XGLM$_{7.5B}$ | (32-shot) | 40.4 | 20.4 | – | **32.1** | **19.4** | **19.8** | **26.3** | **10.6** | 2.4 | 5.9 | 14.5 | **13.7** | 9.7 | 6.6 | **17.1** |
| | mT5$_{3.7B}$ + Sap | (2-shot) | 38.0 | 19.2 | – | 26.7 | 13.7 | 18.3 | 23.5 | 8.6 | **9.2** | 9.9 | 15.0 | 12.1 | **10.8** | 9.7 | 16.5 |
| | mT5$_{3.7B}$ + Sap | (zero-shot) | 38.1 | **21.1** | – | 30.1 | 12.9 | 18.1 | 22.3 | 8.7 | **9.2** | 11.1 | 15.7 | 11.0 | 9.6 | **11.1** | 16.8 |
| ca | Supervised | | 33.4 | 24.8 | 35.1 | – | 19.0 | 21.1 | 28.6 | 15.1 | 13.9 | 13.4 | 18.7 | 20.5 | 2.1 | 2.6 | 19.1 |
| | GPT-3$_{6.7B}$ | (32-shot) | 40.2 | 18.6 | 31.4 | – | 7.0 | 9.3 | 4.3 | 8.0 | 0.9 | 0.9 | 0.3 | 0.4 | 0.1 | 0.1 | 9.3 |
| | XGLM$_{7.5B}$ | (32-shot) | **41.1** | 18.9 | **33.8** | – | 11.3 | 3.3 | **23.9** | **10.8** | 1.3 | 0.8 | 13.8 | 6.1 | 7.9 | 3.1 | 13.6 |
| | mT5$_{3.7B}$ + Sap | (2-shot) | 33.4 | 14.9 | 29.5 | – | 10.7 | 14.4 | 19.7 | 7.0 | 5.6 | 12.4 | 7.3 | **8.7** | **8.7** | 6.7 | 13.3 |
| | mT5$_{3.7B}$ + Sap | (zero-shot) | 37.1 | **19.3** | 32.4 | – | **12.4** | **16.7** | 19.1 | 7.9 | 7.4 | 8.5 | 14.5 | 9.4 | 8.3 | **9.8** | 15.6 |
| fi | Supervised | | 27.2 | 23.0 | 29.3 | 21.6 | – | 20.6 | 26.4 | 16.0 | 14.8 | 12.4 | 14.2 | 19.8 | 1.7 | 0.9 | 17.5 |
| | GPT-3$_{6.7B}$ | (32-shot) | 25.3 | 13.5 | 17.1 | 10.0 | – | 6.4 | 2.8 | 5.7 | 0.7 | 0.7 | 0.3 | 0.3 | 0.1 | 0.0 | 6.4 |
| | XGLM$_{7.5B}$ | (32-shot) | **29.2** | **17.4** | 22.2 | 17.0 | – | **16.5** | **17.5** | **12.4** | 7.5 | 7.6 | 8.0 | 10.1 | 6.2 | 2.0 | **13.4** |
| | mT5$_{3.7B}$ + Sap | (2-shot) | 24.1 | 16.1 | 19.8 | 14.9 | – | 14.2 | 17.0 | 7.0 | 5.8 | 7.1 | 8.3 | 5.6 | **8.5** | **3.9** | 11.7 |
| | mT5$_{3.7B}$ + Sap | (zero-shot) | 23.2 | 16.1 | 20.5 | 16.3 | – | 14.5 | 16.3 | 8.0 | 5.9 | 6.3 | 10.0 | 7.5 | 5.9 | 8.2 | 12.2 |
| ru | Supervised | | 27.5 | 23.5 | 30.1 | 22.0 | 19.4 | – | 31.0 | 16.5 | 15.3 | 13.5 | 18.1 | 20.9 | 2.2 | 2.3 | 18.6 |
| | GPT-3$_{6.7B}$ | (32-shot) | 28.1 | 14.8 | 20.4 | 13.1 | 5.4 | – | 7.4 | 1.2 | 0.2 | 0.2 | 0.2 | 0.1 | 0.1 | 0.1 | 7.0 |
| | XGLM$_{7.5B}$ | (32-shot) | 30.4 | 17.9 | 24.0 | 14.6 | 8.0 | – | **26.3** | **11.6** | 5.5 | 7.4 | 7.1 | 9.1 | 7.3 | 3.1 | 13.2 |
| | mT5$_{3.7B}$ + Sap | (2-shot) | 26.9 | 16.6 | 22.4 | 14.5 | 11.2 | – | 25.2 | 6.1 | 8.0 | 6.4 | 11.3 | 9.1 | **9.8** | 8.4 | 13.5 |
| | mT5$_{3.7B}$ + Sap | (zero-shot) | 27.9 | 17.1 | 22.5 | 19.4 | 13.1 | – | 25.4 | 8.3 | **8.7** | 9.1 | 12.0 | 9.0 | 9.0 | **10.3** | 14.8 |
| bg | Supervised | | 33.0 | 26.1 | 33.7 | 24.9 | 20.8 | 26.5 | – | 17.5 | 16.4 | 14.5 | 20.9 | 23.1 | 2.3 | 2.4 | 20.2 |
| | GPT-3$_{6.7B}$ | (32-shot) | 21.6 | 11.4 | 16.0 | 9.7 | 4.3 | 6.5 | – | 1.2 | 0.2 | 0.2 | 0.1 | 0.2 | 0.1 | 0.1 | 5.5 |
| | XGLM$_{7.5B}$ | (32-shot) | 35.5 | 19.2 | 26.3 | 12.9 | 14.2 | 22.9 | – | 11.9 | 6.8 | 9.2 | 9.4 | 7.5 | 3.2 | 1.0 | 13.9 |
| | mT5$_{3.7B}$ + Sap | (2-shot) | 31.0 | 17.0 | 23.8 | 18.3 | 10.9 | 22.9 | – | 7.2 | 8.3 | 8.1 | 11.7 | 7.4 | **9.5** | 6.6 | 14.1 |
| | mT5$_{3.7B}$ + Sap | (zero-shot) | 32.5 | 17.3 | 24.5 | **21.7** | 10.6 | **23.2** | – | 8.7 | 7.5 | 9.0 | 13.0 | 8.6 | 7.9 | **10.1** | 15.0 |
| zh | Supervised | | 20.9 | 17.6 | 24.3 | 17.4 | 16.0 | 17.2 | 22.1 | – | 15.9 | 11.6 | 15.5 | 18.5 | 1.9 | 2.5 | 15.5 |
| | GPT-3$_{6.7B}$ | (32-shot) | **21.1** | 9.5 | 14.3 | 8.2 | 4.3 | 3.6 | 1.3 | – | 1.1 | 0.4 | 0.2 | 0.2 | 0.1 | 0.0 | 4.9 |
| | XGLM$_{7.5B}$ | (32-shot) | 20.7 | 8.3 | 8.5 | 10.5 | 4.4 | 4.8 | **14.8** | – | 9.3 | 4.2 | 5.6 | **12.0** | 8.6 | 6.2 | 9.1 |
| | mT5$_{3.7B}$ + Sap | (2-shot) | 19.0 | **10.9** | **14.9** | 11.9 | 8.0 | 10.6 | 11.9 | – | 8.9 | 6.0 | **9.1** | 8.0 | **10.0** | 7.6 | 10.5 |
| | mT5$_{3.7B}$ + Sap | (zero-shot) | 18.5 | **10.9** | 14.8 | **12.8** | 8.8 | **10.7** | 11.8 | – | 9.2 | **6.5** | 9.0 | 8.9 | 8.2 | **8.9** | 10.7 |
| ko | Supervised | | 20.9 | 16.7 | 22.1 | 16.5 | 14.9 | 15.5 | 21.1 | 15.7 | – | 10.6 | 15.1 | 18.7 | 1.9 | 4.0 | 14.9 |
| | GPT-3$_{6.7B}$ | (32-shot) | 8.3 | 4.6 | 6.4 | 4.4 | 2.1 | 1.7 | 0.8 | 2.5 | – | 0.2 | 0.1 | 0.1 | 0.1 | 0.1 | 2.4 |
| | XGLM$_{7.5B}$ | (32-shot) | 19.9 | **10.3** | 13.7 | 5.3 | 1.4 | 1.2 | 10.9 | **11.9** | – | 2.7 | 3.2 | 1.0 | 2.2 | 1.4 | 6.5 |
| | mT5$_{3.7B}$ + Sap | (2-shot) | 18.3 | 10.1 | 13.7 | 11.3 | **7.9** | **10.1** | **12.6** | 7.8 | – | **6.3** | 7.2 | 6.6 | 2.6 | 4.7 | 9.2 |
| | mT5$_{3.7B}$ + Sap | (zero-shot) | 18.1 | 10.1 | **13.8** | **12.8** | 7.8 | 9.9 | 11.4 | 7.6 | – | 5.5 | **8.0** | **6.7** | **8.1** | **8.2** | **9.8** |
| ar | Supervised | | 25.5 | 18.7 | 25.7 | 18.9 | 15.6 | 17.8 | 23.8 | 13.1 | 13.3 | – | 15.4 | 19.4 | 1.8 | 0.9 | 16.1 |
| | GPT-3$_{6.7B}$ | (32-shot) | 10.5 | 5.3 | 9.6 | 6.0 | 2.2 | 2.2 | 0.9 | 0.9 | 0.1 | – | 0.1 | 0.1 | 0.2 | 0.0 | 2.9 |
| | XGLM$_{7.5B}$ | (32-shot) | **27.7** | **12.2** | 17.9 | 8.8 | **8.5** | 9.1 | **18.4** | **8.9** | 0.8 | – | 7.7 | 7.8 | 3.4 | 3.7 | 10.4 |
| | mT5$_{3.7B}$ + Sap | (2-shot) | 23.7 | 10.8 | 17.5 | 11.0 | 8.0 | 12.2 | 13.8 | 5.9 | 7.1 | – | 10.3 | 8.0 | 8.0 | 8.0 | 11.1 |
| | mT5$_{3.7B}$ + Sap | (zero-shot) | 26.9 | 11.5 | **19.8** | **15.9** | 7.8 | **14.5** | 13.6 | 6.3 | 7.6 | – | **11.0** | **8.0** | **8.8** | **9.3** | **12.4** |
| sw | Supervised | | 30.4 | 19.4 | 26.7 | 20.1 | 15.6 | 17.6 | 23.8 | 13.2 | 12.2 | 12.0 | – | 19.2 | 2.1 | 4.0 | 16.6 |
| | GPT-3$_{6.7B}$ | (32-shot) | 5.0 | 2.9 | 3.9 | 2.8 | 1.7 | 1.8 | 1.3 | 1.3 | 0.5 | 0.5 | – | 0.4 | 0.1 | 0.1 | 1.7 |
| | XGLM$_{7.5B}$ | (32-shot) | **31.6** | 13.4 | **21.8** | 15.4 | **10.2** | 13.1 | 15.2 | **9.5** | 6.0 | **8.9** | – | 7.6 | 3.4 | 1.4 | **12.1** |
| | mT5$_{3.7B}$ + Sap | (2-shot) | 27.0 | 12.6 | 19.0 | 15.1 | 9.2 | 12.2 | 15.8 | 5.9 | 6.0 | 8.3 | – | 6.5 | **5.4** | 6.0 | 11.5 |
| | mT5$_{3.7B}$ + Sap | (zero-shot) | 30.0 | **13.5** | 20.0 | **18.0** | 9.5 | **14.5** | 15.8 | 6.4 | 5.7 | 7.7 | – | 6.5 | 2.7 | **7.0** | **12.1** |
| hi | Supervised | | 27.9 | 19.4 | 25.9 | 18.9 | 15.7 | 16.9 | 23.9 | 13.5 | 13.9 | 12.2 | 16.8 | – | 2.5 | 3.8 | 16.2 |
| | GPT-3$_{6.7B}$ | (32-shot) | 1.2 | 0.9 | 1.4 | 0.8 | 0.4 | 0.4 | 0.3 | 0.2 | 0.1 | 0.1 | 0.1 | – | 0.1 | 0.2 | 0.5 |
| | XGLM$_{7.5B}$ | (32-shot) | 25.2 | 12.3 | 15.4 | 13.0 | 8.1 | 9.8 | 11.3 | **10.8** | **8.5** | 6.1 | 4.7 | – | 1.5 | 1.9 | 9.8 |
| | mT5$_{3.7B}$ + Sap | (2-shot) | 25.7 | 12.4 | 17.0 | 13.0 | 8.0 | 12.2 | **15.4** | 7.2 | 4.4 | 7.4 | **8.9** | – | 9.6 | 9.0 | 11.6 |
| | mT5$_{3.7B}$ + Sap | (zero-shot) | **27.1** | **12.6** | **17.3** | **14.3** | **9.0** | **12.4** | 14.5 | 4.0 | 6.7 | **8.1** | **8.9** | – | **10.2** | **12.8** | **12.5** |
| my | Supervised | | 10.0 | 6.9 | 10.4 | 8.5 | 6.0 | 6.7 | 9.5 | 5.7 | 6.1 | 4.6 | 7.2 | 9.1 | – | 2.5 | 7.2 |
| | GPT-3$_{6.7B}$ | (32-shot) | 0.5 | 0.3 | 0.4 | 0.4 | 0.2 | 0.1 | 0.2 | 0.0 | 0.0 | 0.0 | 0.0 | 0.2 | – | 0.1 | 0.2 |
| | XGLM$_{7.5B}$ | (32-shot) | 14.1 | 7.6 | 10.1 | 3.8 | 5.7 | 7.1 | 8.9 | **7.1** | **6.9** | 3.6 | 3.5 | **8.9** | – | 2.6 | 6.9 |
| | mT5$_{3.7B}$ + Sap | (2-shot) | **16.8** | 8.5 | **12.9** | 11.0 | 6.7 | 6.1 | 9.2 | 5.2 | 2.9 | **5.0** | 8.0 | 7.0 | – | 5.7 | 8.1 |
| | mT5$_{3.7B}$ + Sap | (zero-shot) | 16.4 | **9.0** | 11.9 | **11.6** | **6.9** | **8.3** | **10.4** | 5.5 | 3.6 | 4.8 | 6.4 | 7.1 | – | **6.2** | **8.3** |
| ta | Supervised | | 8.3 | 4.9 | 6.8 | 5.8 | 5.0 | 4.7 | 7.0 | 2.5 | 2.3 | 1.1 | 5.2 | 6.9 | 1.2 | – | 4.8 |
| | GPT-3$_{6.7B}$ | (32-shot) | 1.0 | 0.5 | 0.8 | 0.5 | 0.2 | 0.3 | 0.3 | 0.1 | 0.2 | 0.1 | 0.1 | 0.2 | 0.0 | – | 0.3 |
| | XGLM$_{7.5B}$ | (32-shot) | 16.3 | 8.4 | 10.3 | 5.1 | **5.2** | 8.1 | 7.6 | **8.1** | 6.2 | 5.4 | 2.8 | 7.2 | 0.9 | – | 7.1 |
| | mT5$_{3.7B}$ + Sap | (2-shot) | 18.7 | 10.4 | 13.7 | 10.9 | 6.3 | 9.8 | 11.6 | 5.2 | 0.7 | 6.5 | 6.0 | 9.3 | 1.8 | – | 8.5 |
| | mT5$_{3.7B}$ + Sap | (zero-shot) | **20.4** | **10.5** | **14.7** | **12.9** | **8.1** | **10.6** | **13.2** | 7.0 | **6.8** | **6.6** | **8.3** | **10.1** | 2.6 | – | **10.1** |
| avg | Supervised | | 26.0 | 20.2 | 26.7 | 20.0 | 16.7 | 18.5 | 24.5 | 14.1 | 13.5 | 11.8 | 16.3 | 19.3 | 2.1 | 2.5 | 16.6 |
| | GPT-3$_{6.7B}$ | (32-shot) | 18.9 | 9.9 | 14.2 | 9.3 | 4.2 | 4.8 | 2.7 | 4.0 | 0.6 | 0.5 | 0.2 | 0.3 | 0.1 | 0.1 | 5.0 |
| | XGLM$_{7.5B}$ | (32-shot) | **28.5** | 14.9 | 20.6 | 14.4 | **10.9** | 12.4 | **18.5** | **10.9** | 5.9 | 6.1 | 8.5 | **9.7** | 5.8 | 3.5 | 12.2 |
| | mT5$_{3.7B}$ + Sap | (2-shot) | 25.8 | 14.1 | 20.2 | 15.6 | 10.0 | 13.7 | 16.9 | 6.9 | 6.8 | 7.4 | 10.5 | 8.5 | **8.1** | **7.5** | 12.3 |
| | mT5$_{3.7B}$ + Sap | (zero-shot) | 27.1 | **15.0** | **20.9** | **18.2** | 10.5 | **14.8** | 17.1 | 7.9 | **7.5** | **8.0** | **11.5** | 9.2 | 8.0 | **9.7** | **13.2** |

Table 6: Few-shot and unsupervised zero-shot machine translation results on FLORES-101 devtest (spBLEU). Source language in rows, target language in columns. GPT-3$_{6.7B}$ and XGLM$_{7.5B}$ use 32 examples from the dev set for few-shot learning. mT5$_{3.7B}$ uses 2 examples from the dev set for few-shot learning. Supervised results correspond to the M2M-124 615M model from Goyal et al. (2021). XGLM$_{7.5B}$ results correspond to the model from Lin et al. (2021). Underline denotes better than supervised, bold denotes best of GPT-3, XGLM, and mT5. spBLEU computed using the implementation from Goyal et al. (2021).

# B  WMT14 Unsupervised Zero-shot Machine Translation

|  |  | English-French | French-English |
|---|---|---|---|
| GPT-3$_{175B}$ | (self-amplified) | **30.0** | **31.8** |
| mT5$_{3.7B}$ + Sap | (self-amplified) | 29.8 | 31.4 |

Table 7: Unsupervised zero-shot machine translation results on WMT14 English-French test set (SacreBLEU) (Bojar et al., 2014; Post, 2018). GPT-3$_{175B}$ (self-amplified) results correspond to the unsupervised zero-shot "GPT-3 (self-amplified)" results from Han et al. (2021) prior to performing distillation, initial backtranslation, and iterative backtranslation which involved unsupervised weight updates. mT5$_{3.7B}$ (self-amplified) is our fully unsupervised zero-shot approach outlined in Section 4 with a 16 prompt ensemble. The SacreBLEU signature used also follows Han et al. (2021): `BLEU+case.mixed+numrefs.1+smooth.exp+tok.intl+version.1.2.20`)

# C  Filtering and Selection Ablation

|  | English-Russian | Russian-English |
|---|---|---|
| Random Selection | 0.0 | 25.5 |
| mT5Score Filtering and Selection | **20.0** | **26.3** |

Table 8: Unsupervised zero-shot machine translation results on FLORES-101 devtest (spBLEU) using mT5$_{3.7B}$ as described in Section 4. In this experiment, we ablate utilizing mT5Score to filter and select the high-quality synthetic examples during bootstrapping over two language pairs, English-Russian and Russian-English. When using random selection, the synthetic parallel examples choosen may be extremely low-quality or non-sensical leading to a 0.0 spBLEU score after self-amplification as shown for the English-Russian language pair.

# D  Prompt Ensemble Size

|  | English-Russian | Russian-English |
|---|---|---|
| Single Prompt | 20.0 | 26.3 |
| 4 Prompt Ensemble | **20.9** | 27.9 |
| 8 Prompt Ensemble | 20.7 | **28.6** |
| 16 Prompt Ensemble | **20.9** | **28.6** |

Table 9: Unsupervised zero-shot machine translation results on FLORES-101 devtest (spBLEU) using mT5$_{3.7B}$ as described in Section 4. In this experiment, we compare utilizing a single few-shot prompt with two synthetic parallel examples to perform the final translation with utilizing an ensemble of 4, 8, and 16 distinct few-shot prompts each with two synthetic parallel examples that generate 4, 8, and 16 translations respectively from which the best translation (by mT5Score) is selected as the final translation over two language pairs, English-Russian and Russian-English.

# E  English-centric Bootstrap Ablation

|  | English-Russian | Russian-Chinese |
|---|---|---|
| Standard bootstrap | 20.9 | 5.8 |
| English-centric bootstrap | **21.2** | **8.3** |

Table 10: Unsupervised zero-shot machine translation results on FLORES-101 devtest (spBLEU) using mT5$_{3.7B}$ as described in Section 4. In this experiment, we ablate utilizing the English-centric bootstrap described in Section 4.3 over two language pairs, English-Russian and Russian-Chinese.

## F   ENGLISH-CENTRIC BOOTSTRAP EXAMPLES

We outline examples of our modified English-centric bootstrapping process for various language pairs below:

- **Example 1** (Russian-English): No change.
- **Example 2** (English-Russian): In step 1, generate Russian-English synthetic examples using a Russian monolingual corpus. Then, reverse the examples to obtain English-Russian synthetic examples.
- **Example 3** (Russian-Chinese): In step 1, for the first three rounds of the bootstrap, generate Russian-English synthetic examples and Chinese-English synthetic examples using Russian and Chinese monolingual corpora. On the fourth and final round, use an English monolingual corpus along with the reversed previous synthetic examples to produce English-Russian and English-Chinese synthetic examples. Since the same English sentences are used to produce both sets, we can align these to form synthetic Russian-Chinese examples. In step 2, we filter examples using the harmonic mean of the two mT5Scores.

## G   ZERO-SHOT PERFORMANCE WITHOUT SELF-AMPLIFICATION

|  | English-Russian | Russian-English |
|---|---|---|
| Standard | 0.4 | 4.6 |
| Bootstrapping and self-amplification | **21.2** | **27.9** |

Table 11: Unsupervised zero-shot machine translation results on FLORES-101 devtest (spBLEU) using mT5$_{3.7B}$ as described in Section 4. In this experiment, we compare the standard zero-shot performance of mT5 with SAP against the zero-shot performance achievable implementing the bootstrapping and self-amplification techniques from Han et al. (2021) with the adaptations described in Section 4.

## H   PROMPTING T5 V1.1 WITH SAP

Careful readers may ask why we use a multilingual model, mT5, to obtain results for the English-only tasks of QA (SQuAD) and summarization (CNN/DailyMail). While a suitable English-only version of T5 could in theory improve performance, we found issues with T5 v1.1's performance. We choose to run SAP with mT5 due to the observation that T5 v1.1 cannot be prompted as easily as mT5, and thus underperforms.

The inputs seen by T5 v1.1 and mT5 during pre-training are of sequence length 512 tokens where multiple spans in the sequence are dropped (Raffel et al., 2020). Therefore, the prompt template we describe in Section 3, would be out-of-distribution from the pre-training inputs since it may have a sequence length shorter or longer than 512 tokens and only contains a single mask instead of multiple masks.

We find that the mT5 model has generalized to sequences shorter and longer than 512 tokens and to sequences that only contain a single mask, while the T5 v1.1 model has not. It is still possible to prompt the T5 v1.1 model with SAP, but requires formulating a prompt constrained to the same in-distribution length of 512.

Due to this complication, we forgo prompting T5 v1.1 in this paper. Since mT5 and T5 v1.1 were trained identically (the same model architecture and hyperparameters), apart from mT5 being pre-trained on the multilingual mC4 dataset instead of the primarily English C4 dataset, we further hypothesize that this difference between T5 v1.1 and mT5 may be an artifact of which checkpoint is selected after pre-training or the length of pre-training (Xue et al., 2021; Raffel et al., 2020).

# I    LIMITATIONS

SAP requires $T$ total forward passes to produce a generation instead of a single forward pass, where $T$ equals the number of words in the generation before reaching a stop token. For example, to produce a translation that has 14 words, SAP requires 14 inferences of the bidirectional model. For tasks with shorter generations with only a few words, such as multilingual question answering, SAP is more practical, especially since it uses fewer parameters. Depending on the size of inference data, SAP as an inference-only prompting technique may be faster and easier to implement than methods that require fine-tuning. While these inferences must be performed sequentially due to the autoregressive nature of the technique, utilizing batching over a test set can still ensure maximum GPU utilization, which is how our experiments were performed. For longer generation tasks, we believe SAP is prohibitively computationally expensive and it likely would not be suitable for use by practitioners directly despite some evidence of improvements in performance. Nevertheless, SAP uncovers an important result: prompt-based learning is an emergent property of bidirectional models. We hypothesize that further research into pre-training objectives and language model design following Wang et al. (2022), Tay et al. (2022), and Soltan et al. (2022) could yield a bidirectional pre-training objective better optimized for few-shot prompting, lifting the requirement to perform multiple forward passes sequentially to generate longer completions.

# J  SURVEY OF OPEN SOURCE LANGUAGE MODELS

| Model | Architecture | Large (>1B params?) | Max Sequence Length during Pre-training | Pre-training Objective |
|---|---|---|---|---|
| *Unidirectional Pre-training Objectives* | | | | |
| **GPT-family models** (GPT-2, GPT-3) (Radford et al., 2019; Brown et al., 2020) | Decoder-only | ✓ | 1024-2048 | Next Token Prediction |
| **EleutherAI-family models** (GPT-Neo, GPT-J, GPT-NeoX) (Black et al., 2021; Wang & Komatsuzaki, 2021; Andonian et al., 2021) | Decoder-only | ✓ | 2048 | Next Token Prediction |
| **XGLM** (Lin et al., 2021) | Decoder-only | ✓ | 2048 | Next Token Prediction |
| **OPT** (Zhang et al., 2022) | Decoder-only | ✓ | 2048 | Next Token Prediction |
| **BLOOM** (BigScience, 2022) | Decoder-only | ✓ | 2048 | Next Token Prediction |
| *Bidirectional Pre-training Objectives* | | | | |
| **BERT-style models** (BERT, RoBERTa, ALBERT, etc.) (Devlin et al., 2019; Liu et al., 2019; Lan et al., 2019) | Encoder-only | ✗ | 512 | Masked Language Modeling; Next Sentence Prediction |
| **BART-style models** (BART, mBART, etc.) (Lewis et al., 2019; Liu et al., 2020) | Encoder-Decoder | ✗ | 512/512 | Token Masking; Token Deletion; Sentence Permutation; Document Rotation; Text Infilling |
| **T5-style models** (T5, mT5, etc.) (Raffel et al., 2020; Xue et al., 2021) | Encoder-Decoder | ✓ | 1024/512 with Relative Position Embeddings | Random Span Corruption |

## K    SELECTED EXAMPLE GENERATIONS

Task: Few-shot Machine Translation (Example #1)

Dataset: FLORES-101 (Arabic → English)

Prompt Template:

```
Translate Arabic to English.
{{examples}}
Arabic:  {{source_text}}
English:
```

Ground Truth:

```
The 802.11n standard operates on both the 2.4Ghz and 5.0Ghz frequencies.
```

Generation (mT5$_{3.7B}$ + SAP):

```
The wireless standard 802.11n operates at the frequency of 2.4 GHz and
5 GHz.
```

Generation (mT5$_{3.7B}$):

```
The 802.11n wireless standard operates at 2.4 and 5.0
```

Commentary:

SAP generates a lengthier and more fluent translation and correctly translates the units of the frequencies. Both generations add the word "wireless" which is used correctly and is likely to appear next to the words "802.11n" and "standard" but does not exist in the ground truth translation.

Task: Few-shot Machine Translation (Example #2)

Dataset: FLORES-101 (Russian → English)

Prompt Template:

```
Translate Russian to English.
{{examples}}
Russian:  {{source_text}}
English:
```

Ground Truth:

```
In 1956 Słania moved to Sweden, where three years later he began work
for the Swedish Post Office and became their chief engraver.
```

Generation (mT5$_{3.7B}$ + SAP):

```
In 1956, Slania moved to Sweden, where he worked for three years for the
Swedish Post Office and became its chief engraved worker.
```

Generation (mT5$_{3.7B}$):

```
In 1956, Slanya moved to Sweden and became...
```

Commentary:

SAP generates a full length translation and more correctly translates "Słania" to "Slania" instead of "Slanya". While the translation without SAP only generates a partial translation, the word "became" indicates the direction of translation would be less close to the ground truth translation than the direction of translation taken by SAP. Notably, SAP produces a relatively high-quality translation, but a common failure mode is displayed in this example. SAP translates "chief engraver" to "chief engraved worker" which is an imperfect paraphrase likely due to an imperfect multilingual alignment of the word "engraver" in the embedding space of the model.

Task: Few-shot Question Answering (Example #1)

Dataset: SQuAD v1.1

Prompt Template:

```
Answer the question based on the following passage.

{{examples}}

Passage:  {{passage}}
Question:  {{question}}
Answer:
```

Passage:

```
In 1874, Tesla evaded being drafted into the Austro-Hungarian Army in
Smiljan by running away to Tomingaj, near Gračac.  There, he explored the
mountains in hunter's garb.  Tesla said that this contact with nature
made him stronger, both physically and mentally.  He read many books
while in Tomingaj, and later said that Mark Twain's works had helped him
to miraculously recover from his earlier illness.
```

Question:

```
Why did Tesla avoid by fleeing Smiljan?
```

Ground Truth:

```
being drafted into the Austro-Hungarian Army
```

Generation (mT5$_{3.7B}$ + SAP):

```
because he was ill and wanted to avoid being drafted into the
Austro-Hungarian Army
```

Generation (mT5$_{3.7B}$):

```
because he was ill and could not leave the country
```

Commentary:

In this example, the grammaticality of the question itself ("*Why* did Tesla avoid by fleeing Smiljan?" vs. "*What* did Tesla avoid by fleeing Smiljan?") has issues. This seems to cause both generations to attempt to answer a "why" style question with "because" instead of a "what" style question. Notably, the answer generated by SAP does eventually reach correct answer where as the the answer generated without SAP hallucinates a fact: "he [Tesla] ... could not leave the country".

Task: Few-shot Question Answering (Example #2)

Dataset: SQuAD v1.1

Prompt Template:

```
Answer the question based on the following passage.

{{examples}}

Passage:  {{passage}}
Question:  {{question}}
Answer:
```

Passage:

```
The Broncos took an early lead in Super Bowl 50 and never trailed.
Newton was limited by Denver's defense, which sacked him seven times and
forced him into three turnovers, including a fumble which they recovered
for a touchdown.  Denver linebacker Von Miller was named Super Bowl MVP,
recording five solo tackles, 2½ sacks, and two forced fumbles.
```

Question:

```
Who won the MVP for the Super Bowl?
```

Ground Truth:

```
Von Miller
```

Generation (mT5$_{3.7B}$ + SAP):

```
Von Miller
```

Generation (mT5$_{3.7B}$):

```
Von Miller, Denver, NFL, NFLPA
```

Commentary:

Without using SAP, the answer generated is correct, but contains hallucinated tokens towards the end of the generation.

Task: Few-shot Summarization (Example #1)

Dataset: CNN / Daily Mail v3.0.0

Prompt Template:

```
Summarize the article.

{{examples}}

Article:  {{article}}
Summary:
```

Article:

```
London (CNN)The Hatton Garden heist, as it will surely come to be known,
was every safe deposit box holder's nightmare, every movie director's
dream.  Thieves using heavy cutting equipment and rappelling gear broke
into the vault of an esteemed 60-year-old safe deposit company in the
heart of London over the past holiday weekend, possibly taking advantage
of as many as four days to rifle through an uncounted number of safe
deposit boxes.  And they reportedly got away with hundreds of thousands
of pounds worth of gems and cash - even, in the educated guess of one
former police official, as much as 200 million pounds, or...[truncated]
```

Summary:

```
Robbers may have taken advantage of a four-day holiday weekend.
Estimates of the value of the items taken rage from hundreds of thousands
of pounds to 200 million pounds.  The heist took place in a historic
heart of London's jewelry business.
```

Generation (mT5$_{3.7B}$ + SAP):

```
The Hatton Garden heist was a nightmare for jewelers and their customers.
They were 'treated like criminals', while officers said they would double
bail for anyone who was arrested before.  They were arrested while on the
sidewalk .
```

Generation (mT5$_{3.7B}$):

```
The latest jewel heist in London took place over the weekend.
```

Commentary:

SAP generates a lengthier generation with multiple sentences summarizing the article.  Some of the facts presented in the generation, however, are not coherent with the article.

Task: Few-shot Summarization (Example #2)

Dataset: CNN / Daily Mail v3.0.0

Prompt Template:

```
Summarize the article.

{{examples}}

Article:  {{article}}
Summary:
```

Article:

```
Sanaa, Yemen (CNN)Al Qaeda fighters attacked a prison in the coastal
Yemeni city of Al Mukallah early Thursday, freeing at least 270
prisoners, a third of whom have al Qaeda links, a senior Defense Ministry
official has told CNN. Khaled Batarfi, a senior al Qaeda figure, was
among the escapees, officials said.  Dozens of attackers took control
of government buildings, including the city's Central Prison, Central
Bank and radio station during the assault early Thursday, according to
officials.  Government troops arrived early Thursday and clashed with the
al Qaeda fighters, and most of the militants fled,...[truncated]
```

Summary:

```
Al Qaeda fighters attack a prison and other government buildings, freeing
many prisoners.  Government troops clash with the fighters, most of whom
flee.  Yemen is descending into chaos as a Shia-Sunni conflict draws in
regional rivals Saudi Arabia and Iran.
```

Generation (mT5$_{3.7B}$ + SAP):

```
Yemen's al Qaeda fighters escape from prison after clash with government
forces.
```

Generation (mT5$_{3.7B}$):

```
Yemen's prisoners escape from al Qaeda fighting in Yemen
```

Commentary:

SAP correctly characterizes the major theme of the article in its summary. Without SAP, the generation devolves in to an incoherent fact ("prisoners escape from al Qaeda fighting") and only a partial summary is generated. The ground truth summary, however, is notably longer and contains multiple sentences, while the summary generated by SAP in this instance is only a single sentence.

## L    RESOURCES

We provide links and citations to resources used in this paper which provide license information, documentation, and their intended use. Our usage follows the intended usage of all resources.

We utilize the following models:

- mT5 (Xue et al., 2021):
  `https://github.com/google-research/multilingual-t5/`
- T5 v1.1 (Raffel et al., 2020; Lester et al., 2021):
  `https://github.com/google-research/text-to-text-transfer-transform`
  `er/`
- T5+LM (Raffel et al., 2020; Lester et al., 2021):
  `https://github.com/google-research/text-to-text-transfer-transform`
  `er/`

We utilize the following datasets:

- FLORES-101 (Goyal et al., 2021):
  `https://ai.facebook.com/research/publications/the-flores-101-evalu`
  `ation-benchmark-for-low-resource-and-multilingual-machine-transla`
  `tion`
- WMT14 (Bojar et al., 2014):
  `https://www.statmt.org/wmt14/translation-task.html`
- XQuAD (Artetxe et al., 2020):
  `https://github.com/deepmind/xquad`
- SQuAD v1.1 (Rajpurkar et al., 2016):
  `https://rajpurkar.github.io/SQuAD-explorer/`
- CNN / Daily Mail v3.0.0 (Nallapati et al., 2016; See et al., 2017; Hermann et al., 2015):
  `https://huggingface.co/datasets/ccdv/cnn_dailymail`

We utilize the following software:

- Transformers (Wolf et al., 2019):
  `https://github.com/huggingface/transformers`
- Datasets (Lhoest et al., 2021):
  `https://github.com/huggingface/datasets`
- SacreBLEU (Post, 2018; Goyal et al., 2021):
  `https://github.com/ngoyal2707/sacrebleu`
- ROUGE (Lin, 2004):
  `https://github.com/pltrdy/rouge`
- BERTScore (Zhang et al., 2019):
  `https://github.com/Tiiiger/bert_score/tree/master/bert_score`
- `langdetect`:
  `https://pypi.org/project/langdetect/`

We estimate the total compute budget and detail computing infrastructure used to run the computational experiments found in this paper below:

- 1x NVIDIA RTX A6000 / 87GB RAM / 4x CPU – 686 hours

