# OpenReview forum: "Bidirectional Language Models Are Also Few-shot Learners"
_ICLR.cc/2023/Conference — ICLR 2023 poster_

### Official Review · Reviewer_iGmv · 2022-10-23

**Confidence:** 4
**Correctness:** 3
**Technical Novelty And Significance:** 2
**Empirical Novelty And Significance:** 3
**Recommendation:** 6

**Clarity, Quality, Novelty And Reproducibility:**

- The paper is well-written and understandable.
- The procedure to adapt se2seq pretrained LMs for prompt-based learning is novel.
- The methodology for boostrapping prompts for unsupervised MT comes from existing working, but an extension in the form of English-centric bootstrapping method has been proposed. This is shown to be useful in some contexts.

Questions

- Table 4/5: Why does T5+LM even lag behind vanilla mT5+SAP? Why does it also perform worse compared to vanilla mT5 on summarization? It is trained on a prefixLM kind of objective, so it does have bidirectional context as well as a causal decoder. An analysis of the same would be useful.

**Strength And Weaknesses:**

Strengths:

- The paper shows that bidirectional models can perform prompt-based learning with the right prompting strategy without the need for any model adaptation. This is an interesting fix for models trained for denoising objectives.
- The paper also proposes an interesting idea for English-centric bootstrapping for unsupervised, though this is somewhat orthogonal to the central contribution of the paper.

Weakness

- The method is applicable to mT5 style models, but the long-term solution is probably to adapt mT5 models with causal LM or prefix LM objectives. This will make the inference more efficient as well.

**Summary Of The Paper:**

In this paper, the authors explore the use of bidirectional models, particularly T5-style models, for in-context learning. The pre-training objectives of bidirectional models do not make it directly amenable to prompt-based learning, but representations from bidirectional models might be better. This paper introduces an iterative procedure (Sequential Autoregressive Prompting) to adapt mT5 for incontext learning at inference time. The results show that with this simple procedure, bidirectional models can also perform prompt-based learning. The results are also competitive with larger decoder-only language models. The major downside is the need to run multiple iterations of generations to complete the generation, thus making the process inefficient. The paper also discusses the strategy applied to the tasks of summarization and question answering.

**Summary Of The Review:**

The paper proposes a quick inference-time fix for adapting seq2seq models trained on denoising objectives for prompt-based learning. This just that seq2seq models can perform incontext learning, but the particular methodology proposed in the paper has a lot of inference-time overhead and alternative proposals of LM adaptation of seq2seq models in concurrent work will eventually be adopted.  Hence, the impact of this work might be limited.

---

> ### Author Response · Authors · 2022-11-10
> **Response to iGmv**
>
> Thank you for your comments.
>
> Weakness:
>
> 1. We agree. We believe the long-term solution will involve something along the lines of mixing unidirectional and bidirectional pre-training objectives (Wang et al., 2022; Tay et al., 2022; Soltan et al., 2022) during pre-training for a more efficient technique than SAP that could be found in the potential future. We discuss this in our "Conclusion" and "Limitations" (Appendix I) sections.
>
> Questions:
>
> 1. For T5+LM's underperformance, T5+LM was actually never meant to be prompted directly, although it can be. Rather, it was meant to be used as a "preprocessing" step before performing prompt-tuning in the prompt-tuning paper by Lester et al. (2021). We mostly compare to it as it is another technique that can enable prompting without *supervised* weight updates (although, it still requires weight updates for the LM-adaptation). In summary, the LM-adaptation helps improve downstream prompt-tuning, but is not quite good enough for prompting the model directly. It's possible more adaptation steps than the 100K Lester et al. (2021) used would improve performance, but it would defeat the purpose of being an "adaptation" and it would just become closer to pre-training a fully unidirectional model the longer adaptation is performed. More information can be found in the LM-adaptation section of our paper under Section 2.2.

---

### Official Review · Reviewer_CyM9 · 2022-10-26

**Confidence:** 4
**Clarity, Quality, Novelty And Reproducibility:** The quality is relatively good.
**Correctness:** 3
**Technical Novelty And Significance:** 2
**Empirical Novelty And Significance:** 2
**Recommendation:** 5

**Strength And Weaknesses:**

**Strength:**
1. The proposed inference-only SAP approach is an interesting way to use the bidirectional LMs’ low-resource transfer ability on generative tasks such as MT, QA and summarization;
2. Combines multiple techniques (e.g. filtering , prompt ensembling, and English-centric bootstrapping) and applies them to perform better unsupervised low-resource MT results;
3. Sufficient experiments and analysis.

**Weakness:**
1. The technical novelty of the paper is somewhat limited, since the SAP prompt technique is just feeding the generated token back the prompt again so as to get the new representation of the encoder.
2. The paper only verifies the effectiveness of the proposed SAP method on the mT5 models, however, claims that the performance improvements come from the Bidirectional LMs, which is not reliable. This part of performance improvement may only come from SAP's effect on mT5, or SAP's effect on the encoder-decoder architecture model. Not only that, the underperforming attempt of running SAP on the T5 1.1 proves the unreliability of this conclusion.
3. Lack of detail analysis on the computational efficiency.

**Summary Of The Paper:**

This paper studies the effectiveness of bidirectional language models (specifically, mT5 model) on low-resource multilingual generative tasks with the proposed Sequential Autoregressive Prompting(SAP) method. The paper presents a representative case study on the low-resource MT task utilizing the proposed SAP and a self-amplification technique. Further, the generalized ability of the proposed SAP method on bidirectional LMs is also validated on few-shot / zero-shot QA and Summarization task.

**Summary Of The Review:**

This paper proposes an inference-only SAP prompting technique for mT5 model to achieve remarkable results on low-resource multilingual generative tasks, including MT, QA and Summarization. The proposed method outperforms unidirectional models (GPT-3, XGLM) within fewer parameters and in-context training examples. Even though the paper presents sufficient experiments and analysis, it does not give strong evidence to support the point that the low-resource performance boost comes from bidirectional pre-training objectives. What’s more, the technical novelty of the paper is kind of limited.

---

> ### Author Response · Authors · 2022-11-10
> **Response to CyM9**
>
> Thank you for your comments. We have revised the paper to address some of your concerns and responded to others.
>
> Weakness:
>
> 1. We agree the technique is relatively simple. However, the T5 architecture was released in 2020, prior to GPT-3, and as far as we the authors know of, no one has discovered its ability to innately generate responses to prompts using in-context learning in similar fashion to models like GPT-3 in the almost 3 years since T5 was developed. GPT-3, being the first model to demonstrate in-context learning, has greatly influenced the research direction of the field with most large language models being trained with relatively similar architecture choices to GPT-3. Therefore, we still believe the contribution is relatively significant and novel, even though relatively simple, as knowledge of T5's architecture / pre-training scheme's ability to perform in-context learning could influence the research direction for LLMs moving forward for the better.
>
> 2. This is a correct and valid concern we try to be upfront about in the paper: "Still, we concede that our results do not conclusively prove bidirectionality explains the difference in performance. Beyond bidirectionality and pre-training objectives, mT5, XGLM, and GPT-3 further differ in architecture, pre-training corpus, and hyperparameters." However, we feel the results shown with mT5 are fairly significant and merit further investigation by future works than can answer the causality behind the performance boost with more confidence. We state, however, those experiments would likely be extremely computationally expensive and out-of-scope for this this work as it would involve pre-training many LLMs from scratch with different architecture choices / pre-training objectives to compare them: "A complete ablation experiment would be computationally expensive, and we leave this as future work." We take your point, however, and agree there are some open questions created by our results that we hope this work motivates a future work to explore and address. As for T5 1.1, we explain that limitation in detail in Appendix H and Appendix J shows available open source models. Due to the lack of other open source models with a encoder-decoder/decoder-only architecture and a bidirectional pre-training objective, we were not able to experiment broadly with other models. It remains something future work may address. However, we feel the results with mT5, on multiple tasks, is enough to demonstrate our primary claim "Bidirectional Models are Also Few-Shot Learners" as even one example of this is enough to show it is possible and the results with mT5 appear quite strong on our tasks.
>
> 3. We do provide some explanation of efficiency in Appendix I (Limitations): "SAP requires T total forward passes to produce a generation instead of a single forward pass, where T equals the number of words in the generation before reaching a stop token. For example, to produce a translation that has 14 words, SAP requires 14 inferences of the bidirectional model." The amount of FLOPs will depend on the particular task, but admittedly, for longer generation tasks this technique is quite expensive and prohibitive. Because of the effiency concerns, we wouldn't recommend this as a practical technique to be used by practitioners, we have now stated this more explicitly in our revision of the Limitations: "For longer generation tasks, we believe SAP is prohibitively computationally expensive and it likely would not be suitable for use by practitioners directly despite some evidence of improvements in performance." Rather, we believe SAP is an important result that can better inform future language model design and training: "Nevertheless, SAP uncovers an important result: prompt-based learning is an emergent property of bidirectional models. We hypothesize that further research into pre-training objectives and language model design following Wang et al. (2022), Tay et al. (2022), and Soltan et al. (2022) could yield a bidirectional pre-training objective better optimized for few-shot prompting, lifting the requirement to perform multiple forward passes sequentially to generate longer completions."

---

### Official Review · Reviewer_ZTjg · 2022-10-28

**Confidence:** 4
**Correctness:** 3
**Technical Novelty And Significance:** 3
**Empirical Novelty And Significance:** 4
**Recommendation:** 8

**Clarity, Quality, Novelty And Reproducibility:**

As stated above, I believe this paper to be of high quality, both from a scientific point of view as well as for its clarity. The proposed approach is relatively simple but, I believe, original. The authors give enough detail to reproduce the experiments.

**Strength And Weaknesses:**

## Strengths

* A clear, well structured paper with a well thought out set of experiments. Most of the questions I could come up with while reading it I found answered in one of the appendices, including an extensive set of ablation studies.
* Translation/generation performance is impressive compared to traditional prompt-based methods on GPT-like models, in spite of the much smaller model size.
* The evaluation is generally strong, with 14 languages being chosen for translation, and the model being also compared to supervised approaches.

## Weaknesses

* The main question I have is: why? I don't think this is spelled out clearly enough in the main paper. Do the authors expect SAP-like approaches to eventually overtake supervised approaches? If so, the questions around computational complexity are central and should be discussed in much more detail in the main section of the paper, including perhaps some FLOPS statistics for the proposed approach and the current SOTA.
* The supervised translation baseline include in Table 2 should perhaps be replaced with a more recent and stronger multilingual model.
* I would definitely challenge the classification of languages such as `ko`, `ar`, `sw` etc. as being low-resource. These are all languages for which at least 1M training sentences can be found (see e.g. Table 1 in <https://arxiv.org/abs/2207.04672>)

Minor: I believe the meaning of "Supervised" in Table 2 is only explained in the Appendix. I would recommend explaining it in the table caption.

**Summary Of The Paper:**

This paper focuses on prompt-based learning applied to bidirectional language models. This class of models doesn't lend itself well to standard prompt-based approaches due to their objectives only training them to fill in short spans of text at most. The approach proposed in this paper is to iteratively prompt such models, feeding back the original prompts concatenated with the first words of previous generations, to enable them to generate much longer sequences in an "autoregressive" manner.

The authors evaluate the method on a translation task, obtaining better performance with mT5 compared to the much larger causal language models GPT-3 and XGLM, and are also not too far off supervised techniques even on low(ish)-resource languages. They further evaluate on QA and summarisation tasks, showing again strong performance against XGLM.

**Summary Of The Review:**

A strong paper proposing a conceptually simple yet effective prompt-based learning approach for models like mT5. Some questions around computational efficiency I believe should be given a bit more prominence.

---

> ### Author Response · Authors · 2022-11-10
> **Response to ZTjg**
>
> Thank you for your comments. We have revised the paper to address some of your concerns and responded to others.
>
> Weakness:
>
> 1. We do not expect SAP (as a prompting-based approach) to takeover supervised approaches (and we generally believe the field does not expect prompting-based approaches to overtake or replace supervised approaches in the near-term). The explanation for why this result is significant we believe is best summed up by the final paragraph of our "Conclusion" section. We do provide some explanation of efficiency in Appendix I (Limitations): "SAP requires T total forward passes to produce a generation instead of a single forward pass, where T equals the number of words in the generation before reaching a stop token. For example, to produce a translation that has 14 words, SAP requires 14 inferences of the bidirectional model." The amount of FLOPs will depend on the particular task, but admittedly, for longer generation tasks this technique is quite expensive and prohibitive. Because of the effiency concerns, we wouldn't recommend this as a practical technique to be used by practitioners, we have now stated this more explicitly in our revision of the Limitations: "For longer generation tasks, we believe SAP is prohibitively computationally expensive and it likely would not be suitable for use by practitioners directly despite some evidence of improvements in performance." Rather, we believe SAP is an important result that can better inform future language model design and training: "Nevertheless, SAP uncovers an important result: prompt-based learning is an emergent property of bidirectional models. We hypothesize that further research into pre-training objectives and language model design following Wang et al. (2022), Tay et al. (2022), and Soltan et al. (2022) could yield a bidirectional pre-training objective better optimized for few-shot prompting, lifting the requirement to perform multiple forward passes sequentially to generate longer completions."
>
> 2. Our results table follows the structure of the results table in the recently published paper by Lin et al. (2021) for direct and easy comparison to their XGLM results. We believe the "Supervised" baseline should be a reasonable performer as it is the official supervised baseline created by the authors of the FLORES-101 dataset. Since our setting is zero-shot and few-shot, we would not claim that our approach would outperform all supervised methods that use unrestricted training data (likely it would not) and was provided only for reference similar to Lin et al. (2021) presenting it for reference alongside XGLM's few-shot results.
>
> 3. Thank you for the suggestion, we originally took the classification from another paper which had those languages grouped into a "Medium-Resource" category and not "High-Resource". We agree with this reviewer's classification of languages and believe those languages should have been classified as "High-Resource" not "Low-Resource". Our revision and Table 2 reflects this new classification now.
>
> 4. Thank you, we have added the explanation of "Supervised" to the Table 2 caption.

---

### Official Review · Reviewer_77VL · 2022-10-31

**Confidence:** 3
**Correctness:** 3
**Technical Novelty And Significance:** 3
**Empirical Novelty And Significance:** 3
**Recommendation:** 8

**Clarity, Quality, Novelty And Reproducibility:**

**Clarity:** The text, motivation, and methodological descriptions are mostly clear. However, I am unclear on some experimental details as mentioned above.

**Novelty/Originality:** The method is novel to my awareness, but somewhat incrementally novel. However, although in hindsight, the approach is technically just a simple modification over normal prompting,  the method still seems moderately effective and serve an interesting purpose (in bringing bidirectional models on par with unidirectional ones at least for the tasks considered), and this hasn't been done in this form before (to my knowledge). Moreover, "simplicity" here can also count as a virtue. So given these factors, the contribution can still be considered significant.

**Reproducibility**: Besides some unclarities, there seems to be enough explication of the resources and methods for it being reproducible although lack of code is not ideal.



**Strength And Weaknesses:**

Strength:

1. The core technique is quite simple and seems effective.

2. The insight that bidirectional models can be utilized effectively for few-shot prompting without weight updates similar to GPT-like models can motivate more research into this direction and towards building of better larger bidirectional models.

Weakness:

1. I am unclear of some of the experimental settings and how much we can conclude from them. Particularly, when is bootstrapping+self-amplification+ensembling are used? Is it used only during translation?

It also seems odd to use those techniques only for zero-shot and not adapt them (or explore possible adaptions) for 2-shot (given the explanation in 4.4)

My main concern is, however, that if XGLM is not using bootstrapping+self-amplification etc. in Table 2 I am not sure how fair of a comparison it is. Perhaps with those techniques XGLM would  exceed mT5 by a high margin.

(In appendix G the additional techniques over SAP seems to make a huge difference. Appendix B does show that mT5 performs on par with a much larger GPT3 on translation when both are self-amplified but still would be curious about XGLM given its multilingual emphasis similar to mT5. However, I acknolwedge the effectiveness of SAP compared to other variants of prompts to mT5)


2. As the paper acknowledge, SAP requires running the whole encoder-decoder architecture in an autoregressive loop. This can be slow. Particularly critical is that caching isn't anymore possible (wherein caching is generally usable to speed up autoregressive decoding in causal decoders), because the encoder is bidirectional and in the autoregressive loop.

-------------------------------

I am increasing the score to 8. While the weaknesses are not completely resolved: re. 1 there are still enough evidence that mT5 can be made to perform on par with undirectional decoders as the authors have clarified. Regarding 2, authors are upfront about it.

There are still some limitations. The conclusions rely only on mT5 but authors are clear about their justification and there are not many other models at a similar large enough scale. In terms of impact, I am not entirely sure how meaningful the result will be - given other works on improving Seq2Seq pre-training is already underway independent of this work and SAP, as it is now, may not be as practical for adoption. However,  I think the paper still brings a neat empirical insight, and sets up a bar with SAP that we can attempt to approximate or overcome. It can also open questions and investigation of why this sort of autoregression over the whole seq2seq setup works so well and if there can be some ways to make to computation easier. There is also a decent amount of experiments in the paper.

Overall, given these factors, I decided on raising the score.

Some (optional) discussion topic the paper may (or may not) add:

Not related to prompt tuning but - [1] and [2] also uses a similar strategy of puting the bidirectional encoder within an autoregressive loop and shows better compositional generalization. This could be worth some discussion and encourage future directions in more closely studying what this form of technique is bringing to the table.

[1] Disentangled Sequence to Sequence Learning for Compositional Generalization - Zheng et al ACL 2022
[2] Recursive Decoding: A Situated Cognition Approach to Compositional Generation in Grounded Language Understanding -Setzler et al. Arxiv 2022

**Summary Of The Paper:**

The paper motivates the interest for making bidirectional language models better few-shot/zero-shot reasoners from natural language prompts/instructions. The paper argues that bidirectional models can potentially have richer representations but the unidirectional training in the causal decoder-only models typically happens to be more amenable to few-shot/zero-shot instruction following. The paper resolves the dilemma by introducing a new prompting technique, SAP, to get bidirectional models like mT5 to do few-shot/zero-shot tasks on par with (or better than) other large causal decoder-only models like GPT3.


In each time step, SAP generates a token using mT5 from a given prompt but then adds the first generated token to the prompt for the next step (in the next time step the whole encoder-decoder is run again with the new prompt).


The paper also adopts several engineering techniques on top of that like bootstrapping with synthetic few-shot samples (sampled and then filtered with unsupervised scorers) and self-amplication (using those synthetic samples as few shots); also prompt ensembling.


Paper shows better or on par results of mT5 with unidirectional models on zero-shot and few-shot translation/QA/summarization.

**Summary Of The Review:**

SAP, a technique to effectively zero-shot/few-shot prompt bidirectional language models, is introduced. Simple method; results look decent. Bidirectional models are shown to have the potential to be on par with or better than unidirectional model on instruction following and in-context learning. However, some experimental comparisons may not be completely fair and inference speed may take a big hit.

---

> ### Author Response · Authors · 2022-11-10
> **Response to 77VL**
>
> Thank you for your comments. We have revised the paper to address some of your concerns and responded to others.
>
> Weaknesses:
>
> 1. Yes, the "bootstrapping + self-amplification + ensembling" is used only in machine translation (as the main case-study task of the paper). The reason we used these techniques was to be able to directly compare to the results by Han et al. (2021), which does use these techniques. As you correctly mention, we do a fair comparison to those WMT14 results in Appendix B. You're also correct that the "bootstrapping + self-amplification + ensembling" results on FLORES-101 are not directly comparable to XGLM's FLORES-101 results from the perspective of a fair comparison, however, our few-shot results that don't use these tricks, still are directly comparable. Our XGLM results are directly pulled from Lin et al. (2021)'s results table, so we did not perform the "bootstrapping + self-amplification + ensembling" tricks on XGLM as it is quite intensive and not central to the larger theme of this paper about prompting bidirectional models and language model pre-training choices. We include what our zero-shot results (with the tricks) would look like on FLORES-101 for 2 purposes: 1) showing Han et al. (2021)'s tricks also work with mT5 + SAP; 2) demonstrating after matching/slightly outperforming XGLM's few-shot results in our 2-shot setting, that we can go even further using the unsupervised tricks from Han et al. (2021) to establish an even stronger SOTA result in the zero-shot setting for FLORES-101.
>
> We try to state all of this in the paper, which we believe is sufficiently clear, but we are happy to take another look if the reviewer feels this is still confusing:
> "To extend our in-depth case study on the machine translation task, we now perform fully unsupervised zero-shot machine translation with SAP and mT5 following the procedure of Han et al. (2021), which uses a self-amplification technique to boost performance. A comparison of zero-shot performance without self-amplification can be found in Appendix G."
>
> The main point here being that mT5, a model that was previously thought to be unpromptable, can achieve these competitive or better results against XGLM, with less data and and a smaller number of parameters.
>
> 2. We agree with the computational efficiency concerns. See our response to Weakness #1 to reviewer Ztjg for a more detailed response. However, we have now stated this more explicitly in our revision of the Limitations: "For longer generation tasks, we believe SAP is prohibitively computationally expensive and it likely would not be suitable for use by practitioners directly despite some evidence of improvements in performance."

---

> > ### Comment · Reviewer_77VL · 2022-11-18
> > **Thank you for the response**
> >
> > I have increased my score and updated the strengths/weakness section where I have detailed my reasons.

---

### Decision · Program_Chairs · 2023-01-20

**Decision:**

Accept: poster

**Justification For Why Not Higher Score:**

limited technical novelty

**Justification For Why Not Lower Score:**

good contribution wrt prompting in mT5 and obtaining high performance.

**Metareview: Summary, Strengths And Weaknesses:**

The paper shows how to use bidirectional encoder-decoder models to perform well in few-shot/prompting settings. The idea is to prompt the models one word at a time, feeding back the original prompts concatenated with the first words of previous generations. They call it the SAP modeling technique, which make mT5 perform at par with GPT3.

The idea is simple, the paper is well written and the reviewers find the contribution to be salient in showing how to get these encoder-decoder models to perform well.

There is of course not much "technical" novelty, but in the world of pre-trained language models, it is not clear if we can expect that much technical novelty from the papers that study zero shot/few shot/prompting settings. The main weakness is identified by authors themselves, which is further highlighted by reviewer CyM9, that one cannot claim that performance improvements come due to "bidirectionality". They only experiment with mT5. The effect may come due to encoder-decoder.

I suggest that the authors try to check SAP's performance on other equally-sized models, if possible. Maybe they will be able to identify whether the result is mT5 specific or specific to an architecture or what.

**Note From Pc:**

if the above contains the word "oral" or "spotlight" please see: "oral" presentation means -> notable-top-5% and "spotlight" means -> notable-top-25%. As stated in our emails, we are disassociating presentation type from AC recommendations